# Innovative Approaches to Semi-Transparent Perovskite Solar Cells

**DOI:** 10.3390/nano13061084

**Published:** 2023-03-16

**Authors:** Pramila Patil, Sushil S. Sangale, Sung-Nam Kwon, Seok-In Na

**Affiliations:** Department of Flexible and Printable Electronics, LANL-JBNU Engineering Institute—Korea, Jeonbuk National University, 567 Baekje-daero, Deokjin-gu, Jeonju-si 54896, Republic of Korea

**Keywords:** perovskite solar cells, semi-transparent, tandem solar cells, building integrated photo-voltaics, efficiency

## Abstract

Perovskite solar cells (PSCs) are advancing rapidly and have reached a performance comparable to that of silicon solar cells. Recently, they have been expanding into a variety of applications based on the excellent photoelectric properties of perovskite. Semi-transparent PSCs (ST-PSCs) are one promising application that utilizes the tunable transmittance of perovskite photoactive layers, which can be used in tandem solar cells (TSC) and building-integrated photovoltaics (BIPV). However, the inverse relationship between light transmittance and efficiency is a challenge in the development of ST-PSCs. To overcome these challenges, numerous studies are underway, including those on band-gap tuning, high-performance charge transport layers and electrodes, and creating island-shaped microstructures. This review provides a general and concise summary of the innovative approaches in ST-PSCs, including advances in the perovskite photoactive layer, transparent electrodes, device structures and their applications in TSC and BIPV. Furthermore, the essential requirements and challenges to be addressed to realize ST-PSCs are discussed, and the prospects of ST-PSCs are presented.

## 1. Introduction

Global energy demands are increasing every day. To fulfill them, fossil fuels are being used at an advancing rate, leading to the vanishing of natural sources as well as to environmental pollution issues. Solar cells, solid-state electrical devices that convert the energy of light directly into electricity via the photovoltaic effect, are one of the promising technologies made to address these issues. There are several types of solar cells, which are used to convert sunlight into electrical energy: monocrystalline silicon solar cells, polycrystalline silicon solar cells, thin-film solar cells, organic solar cells and hybrid solar cells. Among them, the crystalline silicon (c-Si) solar cell has more than 90% of the market share [1,2]. Germanium-based solar cells use germanium as the semiconductor material instead of silicon; they are more efficient at converting sunlight into electricity, making them suitable for space applications due to their higher temperature tolerance and radiation resistance [3,4]. However, their higher cost limits their use in terrestrial applications.

Perovskite solar cells (PSCs) have drawn a lot of consideration due to rapid increases in power conversion efficiency in a short period, and their current efficiency has reached up to 25.7%, which is equivalent to that of silicon solar cells [5]. The PSC is a PV solar cell that uses organic–inorganic hybrid perovskite (OIHP) as a photoactive layer. OIHP has the same crystal structure as that of the mineral calcium titanium oxide (CaTiO_3_); in general, these compounds are represented by the chemical formula ABX_3_, where ‘A’ represents the monovalent cation (e.g., methyl-ammonium, formammidinium or cesium), ‘B’ denotes a rare divalent metal (Pb^−^ or Sn^−^), and ‘X’ is the halide anion (I^−^, Br^−^, or Cl^−^) that bonds to both [6,7,8,9]. In addition to being highly efficient, the advantage of perovskite solar cells is their easily tunable band-gap. Their band-gap can be easily modified to achieve broad band-gap perovskite for semi-transparent applications by changing the components’ locations at the ‘A’ site and the concentration of halide elements at ‘X’ [10,11,12,13]. In addition, the thermochromic behavior of perovskite, i.e., the characteristic of changing color according to temperature, enables various color implementations [14]. These unique features not only create translucent perovskite layers of various colors but also provide extensions to smart windows, tandem solar cells (TSC) and building-integrated photovoltaics (BIPV).

The semi-transparent perovskite solar cell (ST-PSC) is a form that best utilizes the characteristics of the PSC. Unlike general PSCs, the ST-PSC is characterized by transmitting a significant amount of visible light while converting solar energy. The main performance factors of ST-PSCs are the average visible light transmittance (AVT), usually referred to as the average value of the transmittance lying in the wavelength range observable to human eyes that is between 370 and 740 nm [15,16], along with their power conversion efficiency (PCE) and applications due to their optical properties. Evaluating the optical properties of the materials to be used in ST-PSCs is essential for developing efficient ST-PSCs. The main optical properties include the AVT, human luminosity factor, color rendering index (CRI), corresponding color temperature (CCT) and transparency color perception. An average AVT of 20 to 30% is required for window applications. The second important parameter is the human luminosity factor, as human eyes differ from a spectrometer. Human eyes are sensitive to green light but less sensitive to blue and red wavelengths. Hence, harvesting the maximum number of photons in these regions could be one way to achieve high transparency with efficiency. CRI refers to the capability of the light source to render the precise object color compared with that of the reference light source. The CRI value range lies between 0 and 100, revealing the ability of ST-PSCs to transmit light with the actual color of the observed light. Higher CRI values refer to a high color rendering capacity, whereas lower values refer to a lower rendering capacity. CCT accuracy stems from the color space standard developed in 1931 known as the CIE xy chromaticity diagram [17].

Depending on the combination between PCE and optical properties, ST-PSCs could be extended to various technologies, such as TSC and BIPV. Furthermore, their wide-ranging applications, including windows, curtain walls, canopies, balustrades and shading, have drawn a lot of interest [18,19]. However, ST-PSCs still do not have high-efficiency characteristics due to various factors that cause tradeoffs between efficiency and light transmittance. For example, fabricating ST-PSCs requires a compromise in film thickness. This leads to drawbacks such as V_OC_ and J_SC_ reductions, in turn reducing the device’s efficiency. In addition, the fabrication of thin perovskite films makes it difficult and intricate to scale-up productions; perovskite thin films can easily make pinholes, which can be a major factor in reducing efficiency and stability. Numerous techniques have been developed to concurrently increase solar cells’ transparency and performance in order to circumvent these limitations, e.g., partially covering perovskites and creating island-shaped structures, fabricating ultra-thin perovskites for restraining light absorption, making discontinuous perovskites or utilizing wide band-gap perovskite (WBG) for enhancing transparency [20]. However, the trade-off between high transmittance and efficiency still makes the development of ST-PSCs difficult. These issues highlight the need to explore current development progress and derive potential requirements and strategies for ST-PSC development.

Therefore, the present review focuses on various potential possibilities to develop ST-PSCs and covers five important areas, including components and applications. In detail, this review provides insights into various aspects of ST-PSCs in which advancements in perovskite materials, charge transport materials and different types of the most exploited electrodes used along with various types of transparent electrodes are discussed. Finally, applications focusing on BIPV and tandem applications are covered in the following sections: (i) perovskite photoactive layers for ST-PSCs, (ii) transparent electrodes for ST-PSCs, (iii) stability of ST-PSCs, (iv) device structure for ST-PSCs and (v) applications of ST-PSCs. Finally, the conclusion and perspectives provide further insight into improvements in ST-PSC for future research. The structure and performance of the representative ST-PSCs and used component material are summarized in the tables for each section.

## 2. Perovskite Photoactive Layers for ST-PSCs

A trade-off occurs between efficiency and light transmittance, and it is important in ST-PSC to find a compromise between high efficiency and transmittance. Interestingly, the transparency of the perovskite layer can be altered by varying the halide element, whereby the increased band-gap can transmit more light in the visible region through the perovskite layer. Representative ST-PSCs with perovskites of different compositions and thicknesses are listed in Table 1. For example, in MAPbI_3−x_Br_x_ (0 ≤ x ≤ 3), the perovskite layer has a constant thickness (300 nm), and the average visible transmittance (AVT) of the perovskite layer increases from 10% to 24% as the bromide content increases from x = 0 to x = 1.5 [21]. This is because the band-gap of perovskites containing more bromides increases, and the wavelength of the light absorption spectrum shifts to blue. The composition of perovskite with iodine only appears to be dark brown, which changes to red with an increase in bromide content and turns to clear yellow when the bromide content is more than 80%, whereby the band-gap is changed from 1.5 eV to 2.3 eV with the ratio of halide ions (I and Br) and cations (MA and FA). Moreover, from the viewpoint of stability, the phase separation issue that occurs when the Br concentration is high should be considered [22,23]. To emphasize the phase instability problem, researchers of another study partly replaced the FA cation in the FAPbI_3_ structure with a Cs cation, and the content of Br was optimized, resulting in stable FA_0.83_Cs_0.17_PbBr*_x_*I_3−x_ [24]. The phase instability was completely eliminated in the iodine-to-bromide compositional range of FA_0.83_Cs_0.17_Pb(I_0.6_Br_0.4_)_3_, which showed a band-gap of around 1.75 eV with high crystallinity. The FA_0.83_Cs_0.17_PbBr*_x_*I_3−x_ was applied to the ST-PSC, achieving up to 15.1% PCE and 12.5% stabilized PCE.

On a similar note, in recent reports, Shi et al. tried to overcome the problem between efficiency and transmittance by utilizing a wide band-gap (WBG) perovskite absorber with a composition of Cs_0.2_FA_0.8_Pb(I_0.6_Br_0.4_)_3_ [25]. The perovskite with different concentrations varying from 0.2 to 1 M were prepared with a balance between iodine and bromine concentrations to maintain a balance between AVT and efficiency (Figure 1a,b). With interfacial engineering and defect passivation, for ST-PSCs, they obtained a high PCE of 14.40% along with an AVT of 38%. In another approach, Tong et al. prepared triple cation perovskite (Cs_0.05_FA_0.8_MA_0.15_PbI_2.55_Br_0.45_) with a band-gap of 1.63 eV, resulting in a steady-state PCE of 18.2% and an average NIR transmittance (ANT) of 75% for ST-PSCs [26]. Furthermore, they achieved an efficiency of 25% through a combination of narrow band-gap perovskite solar cells and all-perovskite tandem cells. In another method, wide band-gap (~2.3eV) FAPbBr_3_ films were fabricated by Halpert et al. with different concentrations of precursor solution from 1.2 M to 0.8 M and 0.4 M, represented as C1, C2 and C3, respectively (Figure 1c and the inset of Figure 1c) [27]. The results demonstrate that the C1, C2 and C3 films showed a PCE of 5.71, 3.25 and 1.86 with high V_OC_, and the corresponding AVTs were 35.6, 42.5 and 49.2%, respectively (Figure 1d). These ST-PSCs with high V_OC_ and optical clarity are useful for wearable lenses and automotive windscreen applications. Similarly, in a recent approach, Yu et al. studied the effect of perovskite compositions with Cs, FA, MA, I and Br combinations. They fabricated perovskite with band-gaps of 1.73 and 1.82 eV and reported ST-PSCs with improved efficiency and stability (Figure 1e,f). The ST-PSC with a thickness in the range of 100–400 nm achieved a PCE between 4.2% and 15.4%, with an AVT in the range of 52.4–20.8%. The devices exhibited a stability of 85% for 1000 h under continuous illumination [28].

A thin layer of perovskite is the most common and comfortable approach for obtaining transparency in ST-PSCs. The AVT of the perovskite layer is changed by varying the layer thickness. More visible light can pass through a thinner perovskite layer, which relies on the absorption coefficient as well. The trapping states found at the conduction band minimum (CBM) and valence band maximum (VBM) in polycrystalline perovskite structures are responsible for absorption coefficient changes [29]. In addition, a smooth, uniform and pinhole-free surface is necessary to attain a high transparency, which leads to increased V_OC_ and shunt resistance [30]. Various techniques have been employed to develop thin perovskite absorbers, such as spin coating with a low concentration of precursor solution and different spinning speeds, vacuum evaporation and vacuum-assisted techniques, additive engineering, thermal-pressed recrystallizing, etc. [31,32,33].

The most common and easy approach is to manage the thickness of a perovskite thin layer through spin coating with a different rotational speed with a low-concentration precursor solution. Jen et al. varied the perovskite film thickness between 140 and 240 nm, optimizing spin coating conditions [34]. As the perovskite thickness decreases, the transmittance increases in the visible region of 500–800 nm, and the corresponding AVT is tuned between 37.5 and 13%, as observed in Figure 2a. In addition, the color properties were also modified based on the color coordinates and film thicknesses, but additional optimization was required to obtain a neutral-colored ST-PSC. The optimization of spinning speed was demonstrated by Guo et al. They fabricated various thicknesses of perovskite layers by varying the spinning speed [35]. They demonstrated that the films prepared with 3 wt% of polyvinyl pyrrolidone (PVP) show enhanced PCE, stability and AVT, and an ST-PSC with 3 wt% of PVP shows a PCE of 5.36% and 34% AVT (Figure 2b,c). In addition, they showed that the device color can be changed to red, yellow, and green through the spin coating of gold fragments (inset of Figure 2c).

In order to obtain a thin layer of perovskite without pinholes, vacuum evaporation techniques have been applied. This method was utilized by Chen et al. to develop a pinhole-free perovskite thin layer of all-inorganic CsPbBr_3_ by using the dual-source vacuum co-evaporation (DVSE) technique (Figure 2d) [36]. In this technique, they co-evaporated PbBr_2_ and CsBr, varying the molar ratio and evaporation rate using a thermal evaporator. The controlled slow evaporation rate led to the formation of bigger grains and pinhole-free morphology and achieved a PCE of 5.98% with an AVT of 90.9% in the wavelength range of 530–800 nm (Figure 2e,f). This DVSE approach was further modified to obtain a uniform and continuous pinhole-free perovskite structure with high transmittance, for which the technique of simultaneously using spinning and vacuuming was applied. Kim et al. fabricated fine, smooth perovskite films using a short spinning and vacuum drying (SSVD) process and increased visual transparency by limiting optical scattering at the perovskite layer [37]. The mean roughness of perovskite was determined to be 6 nm, resulting in suppressed scattering of 2% and high visual transparency, yielding 11.0 and 8.7% PCE from the anterior and posterior incidence, respectively. Recently, Faheem et al. demonstrated all-inorganic perovskite with Erbium (Er) passivation, which yielded a PCE of 11.61% [38].

**Figure 2 nanomaterials-13-01084-f002:**
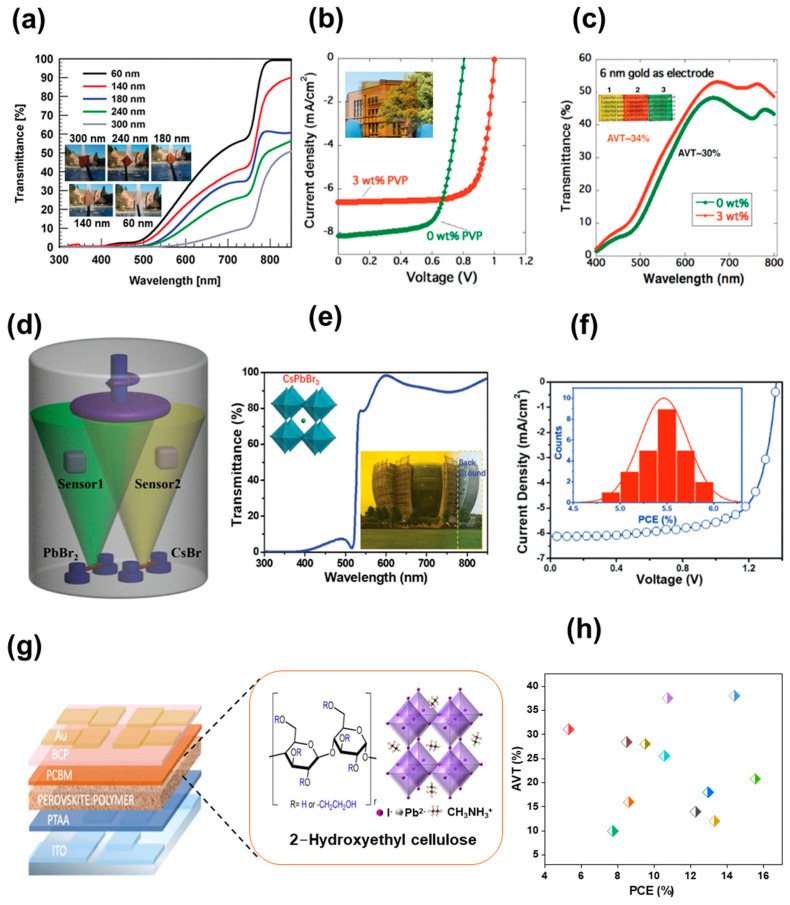
(**a**) UV-Visible spectra with different perovskite film thicknesses (Inset: Photograph of perovskite films with different thicknesses). (**b**) Current–voltage characteristics of the device fabricated with and without PVP. (Inset: Digital photograph of semi-transparent cell.) (**c**) Transmittance spectra for reference device and 3% PVP devices. (Inset: Digital image of various pigment materials coated on Au surface for tuning the color). (**d**) Schematic diagram of the vacuum chamber for the DSVE method. (**e**) Transmittance of the vacuum-evaporated 450 nm thick CsPbBr3 film. (**f**) Current–voltage characteristics of the fabricated semi-transparent cell. (**g**) Schematic of device architecture and 2-Hydroxyethyl cellulose molecular structure and a sketch of MAPbI_3_. (**h**) Interrelation between PCE and AVT for ST-PSCs with different precursor concentrations [21,25,28,30,31,33,34,39,40,41,42,43]. Reprinted with permission from Refs. [34,35,36,44].

Taking into consideration improving transmittance and stability simultaneously, another effective approach is additive engineering. For instance, Zhang et al. introduced the bifunctional additive 1-propyl-[4,4′-bipyridin]-1-ium iodide (BiPy-I) in a perovskite precursor and improved not only efficiency and stability but also transmittance [45]. The large pyridine group in the additive enhances the device’s transparency, and the iodine anions passivate the defects due to the loss of iodine. Consequently, ST-PSCs with an 11.74% PCE and 23% AVT were obtained. On the same line, Bisconti et al. improved the transparency of ST-PSCs by making a blend of perovskite with polymer (Hydroxyethyl cellulose, HEC), as seen in Figure 2g [44]. The ST-PSC with HEC showed an 11.6% PCE with an increased ~44% AVT and obtained a maximum light utilization efficiency (LUE) of 2.4%, which indicates that ST-PSCs with HEC are suitable for window applications. Recently, a new approach was established to produce ultra-thin and superior-quality perovskite layers utilizing a thermal-pressed recrystallizing (TPR) technique [46]. In this technique, the top of the precursor layer is covered using a silicon wafer and is tightly wrapped using a Teflon membrane, and then the packed films are transferred into a hot-press autoclave filled with simethicone under a nitrogen atmosphere. The ST-PSC manufactured with the TPR method demonstrated improvement in overall device performance, including transmittance and stability. The devices showed a PCE of 13.45% after TPR treatment, signifying the capability for the development of ST-PSCs for window applications. The compiled graph (Figure 2h) [21,25,28,30,31,33,34,39,40,41,42,43] depicts the PCE and AVT graph for an ST-PSC modified with different perovskite precursors. The graph clearly indicates the trade-off relationship between PCE and transmittance. The best PCE obtained with good visible transmittance is still 14.4%, which remains low for practical applications. More work and optimization are required to obtain a high PCE with transmittance for practical applications with ST-PSC.

## 3. Transparent Electrodes for ST-PSCs

Transparent electrodes (TEs) are indispensable components that implement ST-PSCs. The basic requirements for TEs are determined by several factors, such as conductivity, chemical stability, cost-effectiveness and, more importantly, good transparency along with well-aligned energy levels with the other layers to minimize the barrier to the transport of charges [47,48]. This is because it is the main role of TE to transmit light and provide an appropriate electric field to collect and transfer the charge.

As shown in Table 2, transparent conductive oxides (TCOs), including fluorine-doped tin oxide (FTO), [49,50], indium tin oxide (ITO) [51], indium-doped zinc oxide [52,53,54] and aluminum-doped zinc oxides [55,56] are usually used as bottom electrodes for ST-PSCs as well as opaque solar cells. These TEs have shown outstanding transparency, low sheet resistance (Rs = 5 to 20 Ω/sq), effective charge collection and long-term stability [18,56,57,58,59,60]. Nevertheless, it is not easy to use these TEs as the top electrodes of the ST-PSC. This is because perovskite is easily damaged during the deposition of the top electrode. Therefore, it is essential to develop a proper deposition method for the top TE to implement high-performance ST-PSCs as well as their excellent electrical and optical properties.

To solve the problems caused by the high-temperature etching of a perovskite layer accompanied by a deposition process such as ITO and FTO, various top electrode deposition methods capable of reducing damage to the photoactive layer, e.g., low-temperature atomic layer deposition or the insertion of a buffer layer, have been developed. Reducing sputtering power is one of the most efficient methods to protect the perovskite layer from damage [50,61]. For example, Bush et al. used the magnetron sputtering method for TCO deposition as the top TE and controlled sputtering power to guard the perovskite layer against damage [56]. In addition, to prevent damage to perovskite, an approach was attempted to coat various robust materials, such as inorganic MoO_x_, AZO nanoparticles, thin-metal layers (Ag and Au), ITO nanoparticles, etc., on the perovskite layer prior to TCO deposition. In previous reports, inorganic MoO_x_ have been used as a buffer layer between TCO and perovskite [52,62,63]. ST-PSCs with the device structure of ITO-coated glass/PTAA or NiO_x_/CH_3_NH_3_PbI_3_/PCBM/AZO/ITO were fabricated by Miyano et al., who made a spin-coated PCBM film and buffer layer (AZO) under sputter-deposited TCO. The AZO layer shielded the PCBM and perovskite films from ITO sputter deposition and eliminated the extraction barrier between PC_60_BM and ITO. As a result, an ST-PSC with a maximum PCE of 12.5% and AVT above 11% was obtained and showed stable performance for up to 4000 h under one-sun illumination [64].

Another problem encountered while using TCO is that conductivity and transmittance are in a trade-off relationship. In general, to increase conductivity, the thickness of the TCO must be increased, but such an increase in thickness leads to a decrease in transmittance. Therefore, increasing conductivity while maintaining transmittance is challenging, and various methods are being studied. For instance, doping has been investigated for the last few years to improve the conductivity of ITO [50]. Zhang et al. used Ce-doped In_2_O_3_ (ICO) as the transparent top electrode to fabricate ST-PSC. The fabricated film had an average of 83.5% transmittance in the 400–1800 nm range, good mobility (51.6 cm^2^/(V·s)) and low resistivity (5.74 × 10^−4^ Ω cm), demonstrating its appropriateness for ST-PSCs. [65] In another approach, Na et al. reported high-performing ST-PSCs made with zinc-doped indium tin oxide (IZTO) as the transparent top electrode, made via linear-facing target sputtering without any added thermal treatment. In this method, they utilized a linear-facing target sputtering system (LFTS) and deposited ITO (10 wt% SnO_2_-doped In_2_O_3_) and IZO (10 wt% ZnO-doped In_2_O_3_) targets at a constant DC power. The resulting IZTO top electrode was utilized for fabricating ST-PSCs and attained a PCE of 12.85%, similar to that of an Ag-electrode-based device of 13.48%. Furthermore, by altering the thickness of the perovskite layer, a PCE of 8.306% was achieved at an AVT of 33.9%. As a result, they suggested an advanced method to increase the conductivity of ITO and reduce damage to perovskite [66]. Furthermore, Tiwari et al. introduced hydrogen into In_2_O_3_ as a dopant, which was applied to ST-PSCs with a particular FTO structure (FTO/ZnO/PCBM/CH_3_NH_3_PbI_3_/Spiro-OMeTAD/MoO_3_/H-doped In_2_O_3_ (In_2_O_3_:H)); In_2_O_3_:H sputtered at room temperature without post-annealing generated a hysteresis-free ST-PSC with good efficiency [50]. In a recent approach, Yoon et al. engineered the top electrode using gallium and titanium-doped indium oxide (IO:GT) in between the top electrode and electron transport layer. They modified the work function to narrow for enhancing the charge transport and reducing the Schottky barrier. The modified devices exhibited a certified PCE of 17.53% with a 21.9% AVT with excellent air stability. Furthermore, four-terminal perovskite–perovskite solar cells exhibited a high PCE of 23.35% [67].

Ultra-thin metal films also serve as an effective approach for TEs, owing to their superior conductivity and transmittance [34,39,68,69,70,71,72,73,74]. Similar to other TEs, it is important to carefully manage the coating thickness to produce appropriate conductivity while maintaining high transmittance for optimal performance, even in thin metal films. Accordingly, a deposition technique using a wet or seed layer for adjusting the thickness of a metal film has been developed; organic and inorganic materials with considerable surface energy, such as polymers [70,75], small molecules [41,71,72,73,74,76,77] and metal oxides [30,76,78,79], have been used for the wet layer and the seed layer. Etgar et al. used the wet deposition method, which created perovskite grids with different dimensions. The transparency of the perovskite film was attained by controlling the perovskite solution concentration and mesh openings. As a result, ST-PSC showed a transparency between 20 to 70% and achieved a PCE of 5% at 20% transparency [68]. For other approaches, instead of a primary metal thin film, to increase the conductivity and transmittance of metal-thin-film-based electrodes, sandwich-structured electrodes using dielectric material/metal/dielectric material (DMD) have been studied [30,37,69,72,74,76,80]. The main principle to utilize DMD structures relies on interference phenomena created by the two dielectric layers to improve transmission through the thin metal film. The DMD TE composed of a MoO_3_–Au–MoO_3_ stack, as seen in (Figure 3a), was utilized by Gaspera et al. to fabricate ST-PSCs [30]. The MoO_3_–Au–MoO_3_ TE showed improved transparency and conductivity, attaining a PCE of 13.6% with a 7% AVT (Figure 3b). On a similar note, Zhao et al. constructed ST-PSCs with SnO_x_/Ag/SnO_x_ as the TE. The SnO_x_ worked as a permeation barrier under the Ag thin film and as a seed layer, inhibiting Ag from reacting with halides in perovskite. In addition, the upper SnO_x_ acted as a capping layer to prevent Ag from reacting to moisture and oxygen and to increase stability. The ST-PSC fabricated with this method showed a PCE higher than 11% with a 70% AVT in the near-infrared region and a 29% AVT, along with excellent stability in an ambient atmosphere with an elevated temperature [69].

Moving further, various metal nanostructures, e.g., AgNWs, CuNWs, Ni mesh, Ag mesh, etc., are employed as TEs for ST-PSCs, owing to their excellent electrical and optical properties and their ability to be applied in low-priced and roll-to-roll processes [41,49,81,82,83,84,85,86]. Among them, AgNW is the most widely used TE material with the easiest application for ST-PSCs. This is because AgNW has considerable advantages as a top TE in ST-PSCs, owing to its excellent electrical and optical properties and simple production procedure [87]. Using this methodology, Brabec et al. constructed an ST-PSC with a structure of ITO/PEDOT:PSS/CH_3_NH_3_PbI_3−_xCl_x_/PC_60_BM/ZnO NP/AgNWs and attained a PCE of 8.49% with an AVT of 28.4% [40]. Here, PCBM and ZnO were applied as energy-level modifiers for better transport and energy level alignment. Pores present in the Ag thin film improve transmittance but can act as a factor that degrades the performance of the ST-PSC. The pores provide a pathway for moisture or oxygen to easily contact and react with the perovskite, which leads to the degradation of the perovskite. Therefore, to address this issue, Goldthorpe et al. employed an encapsulation layer on the devices to prevent damage caused by oxygen and water [88]. Dai et al. utilized an ultra-thin transparent Au layer (UTA) underneath spin-coated AgNWs to physically separate it from direct contact with the active layer (Figure 3c,d). By doing so, they resolved the stability issues and secured ST-PSCs with an 11% PCE, which were illuminated from the front side [81]. Furthermore, Jiang et al. introduced C_60_ as a filler in the Ag NWs network; they revealed that C_60_ can improve energy level alignment and ohmic contact within the buffer layer and AgNWs, achieving stability and an enhanced PCE of 11.02% [89].

Graphene, a two-dimensional carbon material with robust physical and chemical properties, transparency, conductivity, stability and flexibility, has drawn considerable attention for photovoltaic devices and is also being explored for ST-PSCs [90,91,92,93,94,95,96]. Graphene modified with PEDOT:PSS was investigated by You et al. to improve transparency and conductivity and attained a PCE of 12.02% [97]. ST-PSCs were also fabricated with stacked CVD graphene (Figure 4a). The perovskite layer thickness was optimized at 150 nm, and the maximum transmittance reached about 50% at 700 nm (Figure 4b and inset of Figure 4b). The ST-PSC fabricated with stacked CVD graphene achieved a PCE of 5.98% through FTO side illumination [97]. Another carbon form, carbon nanotubes (CNT), are beneficial for creating transparent electrodes in solar cells due to their superior conductivity, flexibility and chemical and mechanical endurance [98,99,100]. However, severe fabrication conditions and solvents limit its use as a top electrode in ST-PSCs [101]; the solvent used in processing needs to be favorable for the active and buffer layer in the device [102]. To overcome this issue, a printing technique was used by Li et al., whereby they utilized carbon grids as top electrodes to produce entirely printable ST-PSCs with a configurable AVT, and the effect of grid spacing was investigated [103]. Grid spacing and PCE have inversely proportional relationships. As grid spacing is reduced, PCE improves and AVT is reduced, and vice versa. Furthermore, the conductivity of transparent carbon grids is improved by inserting a thin layer of multi-walled CNTs, resulting in an 8.21% PCE and 24% AVT. Recently, Seo et al. doped MoO_x_ on a CNT, enhancing its energy-level alignment and hole-transfer rate. Furthermore, the optimized thickness of the MoO_3_ (8 nm)–CNT electrode exhibited 70% transparency at 550 nm, obtaining a PCE of 17.3% [104].

The conductive polymer PEDOT:PSS is well known as a hole-transporting layer for organic solar cells and PSCs [105,106,107,108]. PEDOT:PSS has high potential as a transparent electrode, owing to its minimal sheet resistance and better optical transparency. However, there is a big hurdle to use PEDOT:PSS as a transparent electrode for ST-PSCs. The PEDOT:PSS is dispersed in water, and if it is coated on perovskite directly, the perovskite layer collapses very quickly from the water. To address this problem, Zhou et al. developed a method called the transfer lamination technique. In this method, PEDOT:PSS films are placed on a transfer medium (such as polydimethylsiloxane) initially and are later shifted to perovskite layers to laminate them [109]. They demonstrated that a ST-PSC with FTO/c-TiO_2_/m-TiO_2_/CH_3_NH_3_PbI_3_/Spiro-OMeTAD/PEDOT:PSS configuration shows a 10.1% PCE with a 7.3% AVT after careful modifications of thickness and other parameters (Figure 4c and the inset of Figure 4c). Furthermore, this technique was modified by Zhang et al.*,* and they fabricated TCO-free ST-PSCs by employing PEDOT:PSS as both the top and bottom electrodes [110]. They fabricated low-temperature and fully solution-processable TCO-free ST-PSCs for tandem and flexible applications. The electrically conductive layers were fabricated via an easy and quick casting or printing method, and the TCO on the top and bottom was replaced by nitric-acid-annealed PEDOT:PSS. The champion device demonstrated an efficiency of 13.9% for ST-PSCs, and the flexible PET and PI devices showed a PCE of 10.3 and 7.3%, respectively, with good mechanical stability. After 1000 bending cycles, the ST-PSC retained >90% of its initial PCE, demonstrating exceptional mechanical robustness. To add to this modification, Hu et al. reported that doping a fluorosurfactant can boost the PCE and AVT by improving the phase separation of the polymer network of PEDOT:PSS [111].

**Figure 4 nanomaterials-13-01084-f004:**
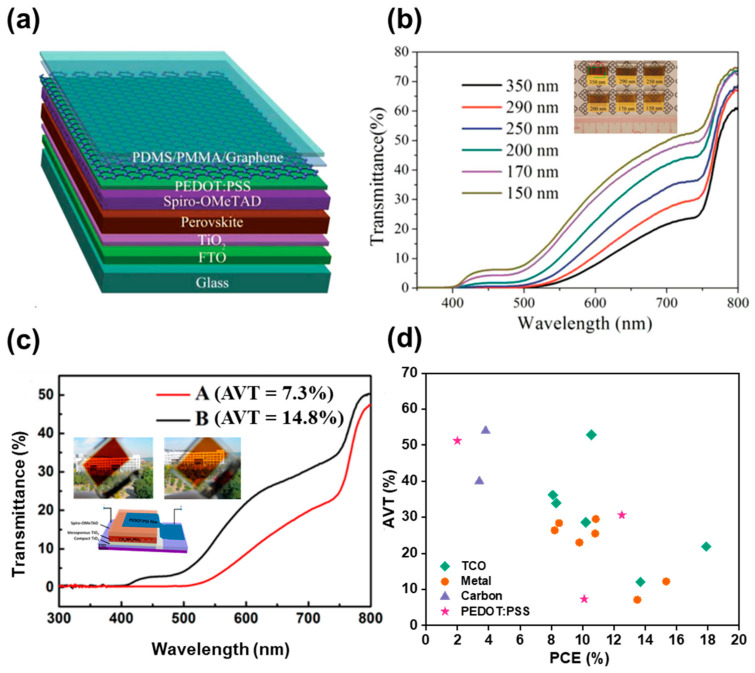
(**a**) Schematic representation of fabricated ST-PSC with graphene as the top electrode. (**b**) Transmittance spectra for the corresponding devices with different thicknesses (Inset: Digital images of cells prepared with different thicknesses). (**c**) Transmittance spectra for cells with different thicknesses (Inset: Schematics of device using PEDOT:PSS and digital images of films with different transparencies). (**d**) Interrelation between PCE and AVT for various electrodes used in ST-PSCs: TCO [64,66,67,112,113]; Metal [40,41,74,114,115,116,117]; Carbon [118,119]; PEDOT:PSS [109,111,120]. Reprinted with permission from Refs. [97,109].

As summarized in Figure 4d, the highest efficiency was achieved in ST-PSCs with TCO. However, the efficiency and transmittance were still low, and efforts to improve them are necessary. On the other hand, although the efficiency was still low, carbon-based electrodes can be a favorable strategy for realizing ST-PSCs with more robustness and high AVT. In addition, in order to overcome the trade-off characteristics of conductivity and transmittance, efforts to improve optical and structural aspects should be continued.

## 4. Stability of ST-PSCs

The stability of ST-PSCs is one of the major bottlenecks for their commercialization. Hence, studies focusing on the stability of ST-PSCs should be carried out in parallel. The heart of solar cells is the active layer, i.e., the perovskite layer, so protecting the perovskite layer from any possible damage is of prime importance. Several efforts have been made to improve efficiency along with stability. However, from a commercialization point of view, a lot of studies need to be carried out to enhance stability without compromising other parameters, such as transmittance and efficiency.

Composition engineering is the most effective and easily accessible way to improve the stability of perovskite solar cells. For example, in perovskite composed of halide ions (Br and I) and cations (MA and FA), their composition ratio greatly affects phase stability. Increasing the Br concentration could improve the transmittance, but it trades off with stability. The phase instability issue is observed when the Br content is increased. Hence, to address this, the Br content was optimized, and the FA cation was partially substituted by a Cs cation. Furthermore, this method was utilized by various researchers to balance the Cs, FA, MA, I and Br combinations and to obtain highly stable devices with improved efficiency and AVT [23,24].

The other approach to improve stability and transmittance simultaneously is the additive engineering method [45]. This method has been utilized by various researchers to improve perovskite precursor quality as well as stability. For instance, some bifunctional group additives enhance the device’s stability and transmittance along with the passivation of defects, contributing to improved efficiency as well. Some researchers have blended perovskite with polymers to obtain smooth, pinhole-free films for improved performance and transmittance. In a recent approach, ultra-thin perovskite layers were fabricated using the thermal-pressed recrystallizing (TPR) technique. This method demonstrates improved performance, transmittance and stability.

The next approach utilizes improvements in the transparent electrodes, which demonstrate a crucial role in ST-PSCs, as the structure of ST-PSCs lies in between two transparent electrodes. Moreover, the main role of TEs is to transmit light and provide an appropriate electric field to collect and transfer the charge. Hence, transparency, conductivity, robustness, cost-effectiveness, energy-level alignment and, most importantly, the mechanical and chemical stability of TEs are of utmost importance for obtaining efficient and stable ST-PSCs. Usually, for bottom electrodes, TCOs such as FTO, ITO and AZO can be utilized. However, the top electrodes are difficult to fabricate, as the perovskite beneath is prone to degradation due to harsh conditions and treatments. Therefore, proper deposition techniques are required for obtaining transparent, conducting and optically transparent electrodes. To avoid damage to the perovskite layer, various methods are adopted, and some of them are low-temperature atomic layer deposition; the insertion of a buffer layer; reducing the sputtering power; and coating with robust materials, such as inorganic MoOx, AZO nanoparticles, thin-metal layers (Ag, Au), ITO nanoparticles, etc., on the perovskite layer before depositing TCO. In another approach, sandwich-structured electrodes using dielectric material/metal/dielectric material (DMD) are used to increase conductivity and transmittance [30]. The fabricated inorganic layer/metal/inorganic layer, in which the bottom inorganic layer acts as a permeation barrier for the metal as well as the seed layer, inhibits metal from reacting with the halide elements in the perovskite layer. Moreover, the top inorganic layer acts as a capping layer to inhibit the metal from reacting with moisture and oxygen, thus improving the ambient and thermal stability of the devices. Furthermore, approaches have utilized AgNWs, along with an ultra-thin Au layer, and few researchers have utilized AgNWs doped with filler materials, such as C_60_, to enhance the stability of ST-PSCs.

## 5. Device Structure for ST-PSCs

Making ST-PSCs with an ultra-thin perovskite layer is the most favorable and comfortable route to achieve efficient solar cells. However, ultra-thin perovskite films can reduce efficiency and cause pinholes, thereby lowering stability. To resolve this issue and to improve the overall visual transparency without sacrificing the performance of the solar cells, the design and pattern of the perovskite layer, the light distribution and the human luminosity component should be taken into consideration. The representative micro- and nanostructured ST-PSCs are summarized in Table 3.

Microstructure design and patterns have been proposed to increase the transparency of ST-PV using transparent regions. This structure has the option to design a neutral-colored ST-PSC by controlling photon absorption areas, including perovskite layers. Microscale templates and selective dewetting techniques have been utilized to fabricate microstructure designs and patterns for ST-PSCs. Eperon et al. first demonstrated the concept of microstructured ST-PSCs defined as “islands” via the partial dewetting of the substrate (Figure 5a) [121]. This interesting property is that the islands were large enough to allow light transmission between two adjacent perovskite islands, but they were tiny enough to appear continuous to the human eye, as seen in the SEM images (Figure 5b) [20]. Thus, the transparency of the device was determined according to the surface coverage and film thickness of the perovskite from 0 to 80%, and the overall device showed a neutral color. In addition, Eperon et al. altered the perovskite structure by substituting FAI for MAI, and they increased the efficiency from 4.9% to 7.4%. However, despite their outstanding color neutrality, the performance of these ST-PSC devices was significantly lower than that of continuous thin films. This was mainly due to insufficient perovskite coverage, as well as the loss of V_OC_, owing to the immediate contact between the ETL and HTL [121]. Horantner et al. controlled a perovskite crystal’s domain size and microstructure using a metal oxide honeycomb structure via colloidal monolayer lithography to overcome this issue [42]. Holes in the honeycomb structure were filled with perovskite, which had a fully controllable domain size and tunable film thickness. The honeycomb zone was primarily transparent; however, the perovskite crystals inside it absorbed enough light. This approach produced ST-PSCs with a PCE of 9.5% and an AVT of roughly 37% in the active layer, and it enhanced the V_OC_ and fill factor. However, this approach requires additional work to optimize pore size and suppress light scattering from rough perovskite surfaces to achieve maximum efficiency. In terms of further improvements, Zhang et al. fabricated ST-PSCs with 30 wt% of perovskite solution, a 600 nm polystyrene scaffold and 8 min of plasma etching, achieving a PCE of 10.3% and AVT of 38% on the same line (Figure 5c,d) [122]. To fabricate patterned and microstructured perovskite films, they used microsphere lithography with a template of a SiO_2_ honeycomb scaffold, derived from a combination of air–water interface self-assembly and oxygen plasma etching. Oxygen plasma etching allowed the precise removal of polystyrene layer-by-layer, which enabled accurate size control of the polystyrene core. These approaches show the feasibility of ST-PSCs, which can be adjusted to a near-neutral color.

Light scattering caused by undesirable optical scattering from the microstructure of ST-PSC can reduce visual transparency and overall device performance. Therefore, light scattering and human luminosity factors should be considered to improve ST-PSCs. It has been reported that nanostructures can improve transparency by reducing surface roughness and can also stabilize the photo-active cubic phase of perovskite [123,124,125,126]. To this end, Kwon et al. used anodized aluminum oxide (AAO) as a scaffold layer for vertically aligned 1D-nanostructure-based ST-PSCs. They controlled the transparency of the perovskite layer via AAO pore size and height. The 1D-nanostructure ST-PSC showed a PCE of 9.6% and a 33.4% AVT, and the self-packing perovskite via the AAO layer improved stability under continuous illumination and reduced hysteresis (Figure 6a) [127]. As another approach to enhance the visual transparency of ST-PSC, Kim et al. considered the human luminosity factor, representing the average spectral discernment for the human visual perception of light. They designed the ST-PSC to absorb photons in blue and red wavelengths, which are less sensitive to human eyes. Photon absorption increases in red regions via plasmon interactions between the TE and Ag nanotube. Consequently, they demonstrated an ST-PSC with a 9.73% PCE and 17.8% AVT via a combination of a perovskite layer that strongly absorbs blue wavelengths and a transparent electrode that improves the absorption of red wavelengths (Figure 6b,c) [128].

## 6. Applications of ST-PSCs

### 6.1. Silicon/CIGS–Perovskite Tandem Solar Cells

As one approach to increase the efficiency of solar cells, a method called tandem that simultaneously uses solar cells optimized for each portion of the solar spectrum has been proposed. The tandem configuration is achieved when two or more photovoltaic systems with different band-gap regions are combined in series or parallel, which, when illuminated by 1 sun, has a strong chance of surpassing the Shockley–Queisser limit of 33.7% for a single-junction solar cell with a band-gap of 1.34 eV [129]. In the tandem solar cell (TSC) with two photovoltaic systems, the upper cell with a wider band-gap absorbs photons with high energy, whereas the bottom side cell with a narrower band-gap absorbs low-energy photons transferred from the upper cell [130]. Thus, in general, the tandem configuration applies the upper subcell with a wider band-gap to harvest higher energy photons, and bottom cells harvest low energy photons.

Based on the device configuration, tandem solar cells are categorized as mechanical/optical four-terminal devices (4T) and monolithic two-terminal (2T) devices [131,132,133,134]. The 4T cells are mainly designed by mechanically stacking, whereby the upper and bottom-side cells are linked through an electrical circuit, which facilitates independency in the fabrication process for the upper and bottom-side cells. These two cells can also be engineered using an optical splitter with high transmission in the longer-wavelength range and high reflection in the short-wavelength range. While in the 2T monolithic tandem configuration, the two subcells are directly connected via an interconnecting layer to form a tandem architect. As there is no electrical circuit, the tunnelling junction and recombination layer perform a crucial role in charge transport and current losses. The corresponding 4T and 2T tandem solar cells are depicted in Figure 7a.

ST-PSCs are interesting options for the top cell in TSCs with c-Si, CIGS and narrow band-gap perovskite solar cells because of their tunable band-gaps [131,135,136,137,138,139,140,141,142]. When selecting acceptable band-gaps, the NIR transmittance of ST-PSCs is critical for attaining highly efficient perovskite-based tandem solar cells (P-TSCs) for both 4T and 2T devices. The optimum band-gap of upper ST-PSCs for achieving the highest PCE for 4T TSCs is 1.8 eV, and that for 2T ST-PSC is 1.75 eV, established in Shockley–Queisser models due to the current limits [143]. Compared to 2T TSCs, 4T TSCs are less susceptible to the band-gap combination of bottom cells. Most P-TSCs are configured with crystalline-Si bottom-side cells because the band-gap of Si (1.12 eV) is well matched with the upper PSC (~1.68 eV) system. Bush et al. delivered the first successful demonstration of a 2T P-TSC, with a certified efficiency of 23.6% for a 1 cm^2^ perovskite–Si tandem solar cell. The perovskite upper cell with a band-gap value of 1.63 eV for a particular composition (Cs_0.17_FA_0.83_Pb(Br_0.17_I_0.83_)_3_) was immediately placed on a smooth Si heterojunction bottom cell with a surfaced back-side [131]. Catchpole et al. stacked ST-PSCs on an inter-digitated back contact (IBC) silicon bottom cell mechanically with a 1.73 eV band-gap perovskite absorber to obtain a tandem efficiency of 26.4% for 4T P-TSCs [144]. Huang et al. employed Cs_0.15_(FA_0.83_MA_0.17_)_0.85_Pb(I_0.8_Br_0.2_)_3_ with a band-gap of 1.64 eV as the active layer and attained a PCE of 25.4% using it for the top in a 2T P-TSC (Figure 7b) [145]. They used the additives MACl and MAH_2_PO_2_ to enhance the perovskite grain size and to passivate grain boundaries; as a result, the top ST-PSC generated a current that matched well with the bottom Si cell. Hou et al. demonstrated a certified efficiency of 25.7% and good device stability with the use of solution processes to make perovskite top cells on textured Si bottom cells [146]. Kim et al. reported an efficient and stable 2T perovskite–Si tandem solar cell with a certified efficiency of 26.2% and an operational efficiency of 26.7%, for which they used anion-engineered additives to increase the stability and performance of perovskite top cells [147]. As a new fabrication approach for realizing R2R, Zhang et al. demonstrated a TCO-free ST-PSC via an entirely vacuum-free and low-temperature solution process [110]. They utilized extremely conductive n-PEDOT:PSS as both the bottom and top transparent electrodes and stacked TCO-free ST-PSCs on the c-Si bottom cell to build a 4T P-TSC, achieving an overall PCE of 19.2%. This approach is not only ideal for creating flexible devices but is also promising for large-scale TSCs in the future.

P-TSCs based on bottom PSCs with a narrow band-gap (~1.2 eV) also have been investigated. Fu et al. reported a 22.1% efficiency using mechanically stacked 4T P-TSCs using narrow-band-gap perovskite and CIGS top cells [132]. Han et al. reported a monolithic two-terminal perovskite–CIGS tandem solar cell (2T P-PSC) with a verified conversion efficiency of 22.43%, which was fabricated with PTAA as the HTL layer, PCBM as the ETL layer and ZnO nanoparticles as a protective layer [148]. Furthermore, a tandem cell composed of only perovskite was studied, in which a narrow band-gap (~1.2 eV) PSC was used as a bottom cell and a wide-band-gap (up to 1.8 eV) PSC was employed as a top perovskite cell. Eperon et al. reported the first perovskite–perovskite tandem solar cell (2T and 4T P-PSC) with a PCE of 17.0% using a FA_0.83_Cs_0.17_Pb(I_0.5_Br_0.5_)_3_ top cell (1.8 eV) and a FA_0.75_Cs_0.25_Pb_0.5_Sn_0.5_I_3_ bottom cell (1.2 eV) [149]. Albrecht et al. employed Cs_0.05_(MA_0.17_FA_0.83_)Pb_1.1_(I_0.83_Br_0.17_)_3_ (1.63 eV) as a perovskite layer for the top 2T P-TSC and attained a TSC with an efficiency of 25.5% in the configuration of a back-side texture mixed with a front-side light management foil [150].

### 6.2. Perovskite–Perovskite Tandem Solar Cells

Based on the tunable nature of the band-gap of perovskite materials, the development of perovskite–perovskite tandem solar cells has been widely studied. The first all-perovskite tandem cell with a 2T configuration was demonstrated by Im et al. They fabricated the device by laminating two different perovskite sub cells of MAPbI_3_ and MAPbBr_3_. The devices yielded high Voc of ~2.25 V, but the fill factor and PCE remained low, with values of 0.56 and 10.4%, respectively, suggesting that tandem cells composed only of WBG perovskite are not desirable [151]. To address this issue, a mixture of WBGs and narrow band-gaps (NBGs) employing a mixture of Sn-Pb perovskites were utilized for fabricating the devices. It was found that the alloying of Pb and Sn can help to narrow band-gap values, and they could be the ideal candidates for bottom cells and could open the way for highly efficient perovskite–perovskites tandem solar cells. The coupling of a bottom Sn-Pb perovskite cell and semi-transparent MAPbI_3_ generated a PCE of 19.08% for 4T devices [152]. Huang and co-workers, via the incorporation of 0.03 mol% Cd^2+^ into a perovskite precursor, de-doped Sn-Pb perovskite and optimized the thickness, yielding an efficiency of 20.2% for single-junction solar cells and 23% for 2T-based tandem devices [153]. Tan et al. utilized a comproportionation reaction to decrease the amount of Sn vacancies in the perovskite precursor by reducing Sn^4+^ to Sn^2+^ by doping Sn into the Sn-Pb mixed perovskite precursor solution. The resulting devices exhibited a PCE of 21.1% and improved fill factor, Jsc and Voc values due to reduced vacancies and an improved charge carrier diffusion length (~3 µm). They also fabricated a 2T monolithic all-perovskite TSC exhibiting a PCE of 24.8% with high Voc. The devices demonstrated good stability as well, retaining 90% stability even after 463 h of MPP under one-sun illumination (Figure 7c) [154]. In a recent study, Hu et al. investigated two novel solution-processed fullerene derivatives as interlayers for Sn-Pb-based narrow-band-gap PSCs. The interlayer improved charge transportation and decreased non-radiative recombination, delivering an improved PCE of 18.6% with improved Voc and Jsc. They also demonstrated 4T all-perovskite TSCs with a PCE of 24.8%, utilizing an ST wide band-gap of 1.63eV and an NBG of 1.26 eV [155]. Furthermore, Li et al. fabricated flexible all-perovskite tandem solar cells with nearly 25% efficiency and with good bending stability. They incorporated a molecular-bridged hole selective contact (HSC) to overcome interfacial recombination and to enhance hole extraction in flexible devices on a low-temperature NiO film. With this strategy, they achieved a PCE of 24.7% and 23.5% for 0.049 cm^2^ and 1.05 cm^2^, respectively.

**Figure 7 nanomaterials-13-01084-f007:**
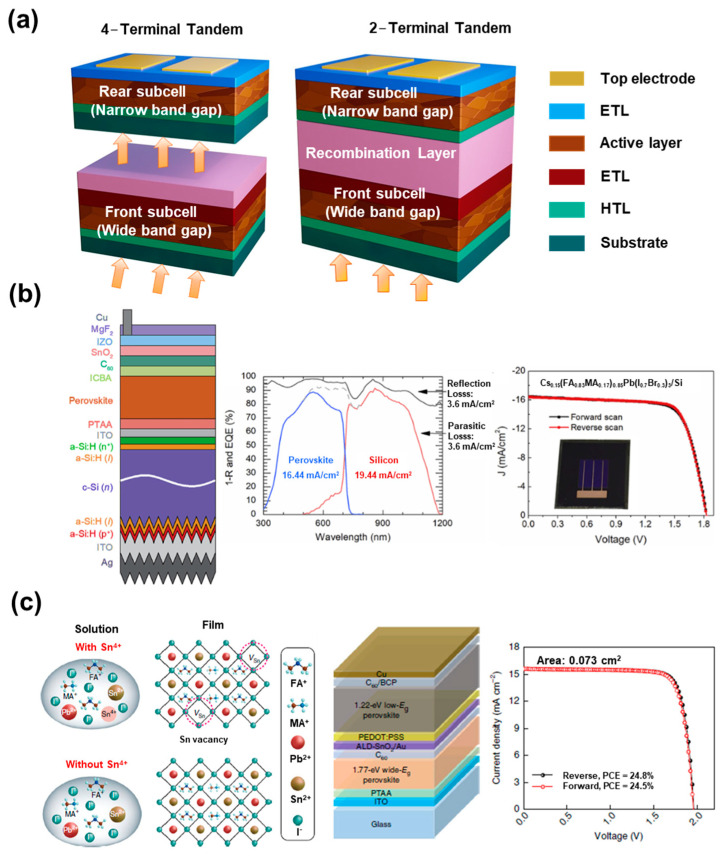
(**a**) Schematic illustration for 4–terminal and 2–terminal perovskite–based tandem structure. (**b**) Schematic structure for perovskite–silicon monolithic tandem devices, J–V curves of perovskite–Si tandem device under forward and reverse scans with an inset photograph of the tandem device, and EQE and total absorbance of perovskite–Si tandem device. (**c**) Schematic of the formation of Sn vacancies in mixed Sn–Pb perovskite along with device architecture and J–V curves of the tandem solar cell. (**c**) Schematic illustration of planar heterojunction PSCs, energy-level diagram for NBG perovskite and different fullerene derivatives, and current density–voltage (J–V) characteristics of champion devices. Reprinted with permission from Refs. [145,154].

Yang et al. used a mixture of Pb and Sn to fabricate a Pb-Sn binary 4T all-P-TSC. Upon increasing the Sn concentration from 0 to 75%, the band-gap of MAPb_1−x_ Sn_x_I_3_ decreased from 1.61 to 1.27 eV [152]. Recently, guanidinium thiocyanate (GuaSCN) was added into a Sn-Pb PVK film by Tong et al. to improve the carrier lifetime. Perovskite with (FASnI_3_)_0.6_(MAPbI_3_)_0.4_, having a 1.25 eV band-gap, demonstrates a carrier lifetime of more than 1 us. In addition, when it is combined with a Cs_0.05_ FA_0.8_MA_0.15_PbI_2.55_Br_0.45_ (Eg = 1.63 eV) front cell, a resultant PCE of 25.4% is obtained for 4T tandem cell [26]. Furthermore, Nejand et al. utilized a vacuum-assisted growth method to control the thin-film formation of Sn-Pb binary perovskite with a band-gap of 1.27 eV. They obtained a PCE of 18.2% with increased charge carrier lifetime. When it was stacked with wide-band-gap perovskite, Cs_0.1_(MA_0.17_FA_0.83_)_0.9_Pb(I_0.83_Br_0.17_)_3_ (Eg = 1.63 eV), it exhibited a PCE of 23% for 4T all-perovskite tandem solar cells [156].

### 6.3. Building-Integrated Photovoltaic (BIPV) Applications

PSCs are evaluated as suitable devices for ST-PV, owing to their high efficiency, excellent transmittance and low-cost manufacturing. Neutral-colored ST-PSCs display potential for fulfilling the requirements of power-generated windows on cars, automobiles and buildings. There are various applications in which ST-PSCs are being used in day-to-day life. The application of ST-PSCs to building-integrated photovoltaics (BIPV) can provide an extended opportunity for the usage of solar energy. A semi-transparent BIPV allows for the replacement of conventional building coverings, such as window panes, roof tops, curtain walls, etc., and, at the same time, it reduces energy demand, controls heat loss and can be used for glazing for comfortable daylight (Figure 8) [157,158].

Several attempts have been made in this regard to develop efficient BIPV systems. Various PV technologies have been reported for BIPV applications, which include amorphous silicon semi-transparent solar cells, Cu(In, Ga)Se_2_ (CIGS) semi-transparent solar cells, dye-sensitized cells (DSCs) and perovskite solar cells [159]. Among them, PSCs are the most promising PV technology for semi-transparent BIPV recently. Two approaches are being tried to increase the transparency of PSCs; research on adjusting the thickness and shape of the perovskite layer for improving transmittance is being conducted. To the human eye, the microstructure or nanostructured perovskite layer appears neutrally colored, with little effect on the spectral properties of light transiting through it.

Cannabale et al. conducted real-time research by applying it to buildings in Bari, Italy, to confirm the efficiency of BIPV technology [160]. The overall results concluded that semi-transparent PV-based devices could be a better choice depending on the glazing and shading conditions. For example, the top floor was more effective than the middle floor or the ground floor; this was because the glare and shading of other buildings can reduce the efficiency of the middle floor and the ground floor. In another modification, Ghosh et al. demonstrated carbon-based ST-PSCs for BIPV applications and achieved an 8.13% PCE with an average solar transmission of 30% and an average visible transmission of 20% [161]. Martellotta et al. developed BIPV using amorphous silica (a-Si) and perovskite [162]. They fabricated ST-PSCs with a 6.64% PCE and 42.4% transparency and compared them with a Si-based BIPV with a 4.80% PCE with 30.1% transparency. These examples confirm that PSC is a suitable candidate for BIPV applications.

Lightweight and flexible solar cells can provide scalability to various applications through integration with BIPV technology, which includes smart buildings, windows, facades, etc. BIPV is being studied based on perovskite and lightweight substrate materials, such as polyethylene terephthalate (PET) and polyethylene naphthalate (PEN). In terms of this modification, Kang et al. fabricated ultralight and flexible ST-PSCs using orthogonal silver nanowires on 1.3 µm thick polyethylene naphthalate foils and achieved a PCE of 15.18% with a power-per-weight of 29.4 Wg^−1^, which is the maximum reported value for lightweight solar cells [163]. For other applications, Yang et al. demonstrated thermochromic solar cells with more than 7% PCE applicable to smart solar windows, in which the transmittance of solar cells changes depending on the temperature [164]. They used the phase transition of CsPbBr_x_I_3−x_ perovskite according to the temperature. The transparent non-perovskite phase has a visible transparency of 81.7%, whereas the dark perovskite phase is 35.04% visibly transparent. Furthermore, Neale et al. developed PV windows with modulated optical properties controlled by solar illumination and achieved a PCE of 11.3% with 68% visible transmittance in the bleached state [165]. The flexibility and lightweight nature of this technology make it easier to integrate flexible modules into advanced building applications [166,167]. The use of various new flexible substrates, such as ultra-thin glass, foldable plastics, textiles, etc., summarizes the advances and prospects in flexible lightweight technology [168]. However, the advantages of flexible substrates are accompanied by prospective challenges as well. To utilize the flexible substrates PET or PEN as TEs, usually a TCO, i.e., indium tin oxide (ITO) or indium zinc oxide (IZO), and sometimes an ultra-thin silver layer is typically coated onto them. Despite their good conductivity and transparency compared to TCO/glass substrates, these substrates are prone to easy damage due to bending, as the coated ITO material is brittle and can be easily damaged. Secondly, these ITO layers are annealed at lower temperatures, which demonstrates their diminished chemical resistance as compared to that of crystalline ITO substrates. Specifically, if the substrates are treated in any acidic solution, that could lead to the degradation of the above-lying perovskite layer if the layer is not covered with a compact charge transport layer. This adds to the thermal constraints for the fabrication of n-i-p-based PSCs, as the fabrication process should be carried out at a temperature below 150 degrees [169]. Hence, the careful modifications of PET substrates with appropriate conducting materials are required to overcome the underlying challenges.

## 7. Conclusions and Perspectives

ST-PSCs are one of the most promising technologies for achieving sustainable and carbon-free energy. In addition, ST-PSCs are suitable for various applications, such as TSCs and BIPV, due to their excellent optoelectronic properties and PCE. Therefore, ST-PSCs have attracted a lot of attention in a wide range of applications, such as windows, curtain walls, canopy, railings, shading, etc. However, ST-PSCs have a challenge due to the compromise between efficiency and light transmittance. To overcome these challenges, many studies are underway, including band-gap tuning, transparent electrodes, and creating island-shaped microstructures. Here, a brief overview of recent progress on ST-PSCs is described, and the essential requirements for realizing ST-PSCs are discussed. The structure and performance of the representative ST-PSCs and each component material are summarized in the tables for each section.

For the realization of ST-PSCs, adding personal views on future development directions, there are some promising directions for ST-PSCs and challenges to be solved. Regarding perovskite absorbance, more effort should be made to develop stable new perovskite with an adjustable band-gap. In addition, along with a deposition technique that can accurately adjust the thickness of the perovskite layer, a design or structure that can control transmittance without performance degradation should be developed. The perovskite layer, with an optically enhanced microstructure or nanostructure, could be one of the promising directions for ST-PSCs. For TEs, various materials are currently being studied for transparent electrodes, and there is the possibility of development in each direction. However, because TEs can determine efficiency and transmittance as well as stability, the following factors should be considered when developing TEs: TEs should be excellent in conductivity, transparency, chemical stability and cost-effectiveness, and the energy level of TEs should be well aligned with the other layers to minimize the barrier for the transport of charges.

The feasibility of ST-PSCs and their future prospects will be brighter. PSCs are one of the most favorable technologies for addressing the climate crisis. PSC power generation capacity is expected to grow above 21.9 TW to achieve carbon freedom by 2050. TSCs, which combine two or more photovoltaic systems with different band-gap regions in series or parallel, have a huge possibility of beating the Shockley–Queisser limit of 33.7% with a band-gap of 1.34 eV under one-sun illumination for a single-junction solar cell [170]. The theoretical PCE of TSC is calculated as ~46% for two junctions (2J), ~50% for three junctions (3J) and 65% or more for an infinite number of junctions, taking into consideration both solar irradiance and electroluminescence by assuming 100% radiative emission of other cells in TSCs. ST-PSCs have a high potential for use as the top cells of TSCs with c-Si, CIGS and narrow-band-gap PSCs due to their adjustable band-gap with good transparency. Silicon solar cells and PSC-based tandem cells have been reported to have an efficiency of close to 30%. In particular, considering that the building sector currently consumes 40% of the planet’s energy and that the figure will be about double to triple by 2050, this increase in photovoltaic (PV) energy is expected to be more evident in the building sector. The application of ST-PSCs to BIPV can provide an extended opportunity for the use of solar-based energy. Based on ST-PSCs, BIPV can replace traditional building envelopes, such as windows, curtain walls, shading, etc. Expanding the total power output of the module per unit of area is the most obvious method to continuously lower the overall price of installed PV power generation and expand the prevalence of PV.

Therefore, the development of high-performance ST-PSCs not only in tandem but also for BIPV is expected to become more active.

**Table 1 nanomaterials-13-01084-t001:** Summary of representative ST-PSCs with different perovskites and charge transport materials.

Category	Band-Gap (eV)	Thickness (nm)	V_OC_ (V)	Jsc (mA/cm^2^)	FF (%)	PCE (%)	AVT (%)	HTL	ETL	Ref.
MAPbI_3-y_Br_y_	1.6–2.3	300	1.02	20.92	66.3	14.15	9	Spiro-OMeTAD	C_60_	[21]
			1.07	17.6	71.9	13.54	11			
			1.11	15.62	70.2	12.26	14			
			1.13	12.79	69.7	10.03	17			
MAPbI_3_	~1.5–1.6	54	0.71	9.7	66	5.3	31	Spiro-OMeTAD	TiO_2_	[30]
		107	0.94	13.7	63	8.8	19			
		141	0.95	14.7	65	10.1	16			
		289	0.98	20.4	58	13.6	7			
MAPbI_3−x_Cl_x_	~1.57–1.59	70	0.97	11.55	72.23	7.81	42	PEDOT:PSS	PC_60_BM/D-ZnO	[31]
		100	0.98	14.23	72.07	9.55	33			
		129	0.97	17.32	69.38	10.81	28			
		339	0.97	19.1	70.85	12.95	18			
		180	1.07	12.2	76	10.22	25.7			
		240	1.06	13	73	10.73	37.5			
MAPbI_3_	~1.5–1.6	310	0.94	18.3	73	13.3	12	PEDOT:PSS	PCBM/C_60_	[39]
		230	0.94	16.4	72	11.8	16			
		180	0.94	14.4	69	9.3	24			
		150	0.94	13.8	67	8.7	29			
		110	0.94	11.7	67	7.4	34			
		65	0.87	7.5	59	3.8	47			
MAPbI_3_	~1.5–1.6	220	0.96	15.87	69.68	10.55	25.5	PEDOT:PSS	ALD-ZnO	[41]
MAPbI_3−x_Cl_x_	~1.57–1.59	150	0.964	13.18	66.8	8.49	28.4	PEDOT:PSS	PC_60_BM/ZnO	[40]
MAPbI_3−x_Cl_x_	~1.57–1.59	-	0.84	17.1	66	9.5	28	Spiro-OMeTAD	TiO_2_	[42]
MA_0.7_FA_0.3_Pb(I_y_Br_1−y_)_3_	1.57	-	1.02	21.47	75	16.42	-	Cu-doped NiO_x_	PC_61_BM:C_60_(1:1) /Bis-C_60_	[171]
	1.6		1.06	20.48	77	16.72	-			
	1.63		1.1	20.21	78	17.34	-			
	1.66		1.11	18.94	78	16.4	-			
	1.69		1.11	17.34	78	15.01	-			
MAPbI_3−x_Cl_x_	~1.57–1.59	240	0.94	14.67	62.34	8.6	15.94	PEDOT:PSS	PCBM	[43]
Cs_0.2_FA_0.8_Pb(I_0.6_Br_0.4_)_3_	~1.57–2.28	362	1.22	15.49	75.96	14.4	38	NiOx-modified with 2PACz	C60/BCP	[25]
Cs_0.05_FA_0.64_MA_0.31_PbI_2.01_Br_0.99_	1.73	400	1.272	16.12	75.83	15.55	20.77	Spiro-OMeTAD	SnO_2_	[28]

**Table 2 nanomaterials-13-01084-t002:** Summary of representative ST-PSCs with different top TEs.

Category	Device Structure	AVT (%)	PCE (%)	Ref.
Transparent conductive oxide-based TEs	FTO/ZnO/PCBM/CH_3_NH_3_PbI_3_/Spiro-OMeTAD/MoO_3_/H-doped In_2_O_3_	-	14.1	[50]
	ITO/PEDOT:PSS/perovskite/PC_60_BM/AZO/ITO	-	12.3	[56]
	ITO/PTAA/CH_3_NH_3_PbI_3_/PCBM/AZO/ITO	12.08	13.68	[64]
	ITO/NiO/perovskite/PCBM/ZnO/IZTO	33.9	8.31	[66]
	ITO/ZnO/PTB7-Th:IEICO 4F/MoO_3_/Ag/ITO	36.2	8.1	[112]
	ITO/ZnO/PM6:Y6:PC_71_BM/MoO_3_/Ag/ITO	28.6	10.2	[112]
	ITO/NiOx/PSS/FAPbBr_0.43_Cl_0.57_/PC_61_BM/ZnO-NPs/LS-ITO/M-PEDOT:PSS/PTB7-Th:6TIC-4F/ZnONPs/ITO	52.91	10.55	[113]
	ITO/NiOx/PSS/Perovskite/PCBM/BCP/IO:GT	21.9	17.9	[67]
Metal-based TEs	ITO/PEDOT:PSS/CH_3_NH_3_PbI_3_/C_60_/BCP/Ag/MoO_3_	7.1	13.49	[74]
	ITO/ZnO/PM6:N3/MoO_3_/Ag/MoO_3_	28.94	10	[172]
	ITO/SnO_2_/FAPbI_3_/spiro-OMeTAD/MoO_3_/Ag/WO_3_	12.18	15.33	[114]
	ITO/NiO/Cs_0.175_FA_0.825_Pb(I_0.875_Br_0.125_)_3_/C_60_/Ag/C_60_	-	5.1	[173]
	ITO/PEDOT:PSS/PTB7-Th:IEICO-4F/PFN-Br/Ag/PCs	29.5	10.83	[115]
	ITO/PEDOT:PSS/PTB7-Th:ITVfIC/PDINO/Ag	26.4	8.21	[116]
	ITO/PEDOT:PSS/perovskite/ALD-ZnO/AgNW/ALD-Al_2_O_3_	25.5	10.8	[41]
	ITO/PEDOT:PSS/CH_3_NH_3_PbI_3−x_Cl_x_/PC_60_BM/ZnO NP/AgNWs	28.4	8.49	[40]
	FTO/TO_2_/CH_3_NH_3_PbI_3_/spiro-OMeTAD/AgNWs–Au	-	11.1	[89]
	ITO/ZnO/PM6:Y6/PEDOT:PSS/AgNW	23	9.79	[117]
Carbon-material-based TEs	FTO/TiO_2_/CH_3_NH_3_PbI_3−x_Cl_x_/spiro-OMeTAD/PEDOT:PSS/graphene	-	6.13	[97]
	PEN/graphene/PEDOT:PSS/ZnO/PDTPDFBT:PC_70_BM/MoO_3_/graphene	54	3.8	[118]
	Graphene/PEDOT:PSS/ZnO/PTB7:PC_71_BM/PEDOT:PSS/graphene	40	3.4	[119]
	ITO/ZnO/PTB7:PC_71_BM/MoO_3_/HNO_3_-CNTs	-	3.7	[174]
	ITO/SnO_2_/MaPbI_3_/CNT/MoO_3_/Spiro-OMeTAD/Au	-	17.3	[104]
PEDOT:PSS-based TEs	FTO/TiO_2_/CH_3_NH_3_PbI_3_/Spiro-OMeTAD/PEDOT:PSS	7.3	10.1	[109]
	ITO/PEDOT:PSS/FAMAPbI_3−x_Br_x_/PCBM/PEDOT:PSS:CFE/PDMS	30.6	12.5	[111]
	ITO/ZnO/PEIE/P3HT:PCBM/PEDOT:PSS	51.2	2	[120]

**Table 3 nanomaterials-13-01084-t003:** Summary of representative ST-PSCs with different micro- and nanostructures.

Feature	Device Structure	V_OC_ (V)	J_SC_ (mA/cm^2^)	FF (%)	PCE (%)	AVT (%)	Ref.
Islands	FTO/TiO_2_/MAPbI_3−x_Cl_x_/Spiro-OMeTAD/Au	0.81	0.71	61	3.5	26.8	[20]
Islands	FTO/c-TiO_2_/MAPbI_x_Cl_3−x_/Spiro-OMeTAD/Ni microgrid	0.92	10	64	6.1	38	[85]
Islands	FTO/TiO_2_/FAPbI_3_/Spiro-OMeTAD/Au	0.86	14.2	60	7.4	33.6	[121]
Islands	FTO/TiO_2_/MAPbI_3_/Spiro-OMeTAD/Au	0.64	10.6	54	4.9	40.5	[121]
Honeycomb	C-TiO_2_/SiO_2_ HC patterned MAPbI_3_ (40%, mass fraction)/Spiro-OMeTAD/Ag	0.98	17.5	0.56	9.8	15	[122]
Nanocubes	ITO/PEDOT:PSS/MAPbI_3_ (180 nm)/PCBM/Ag NCs/BCP/Ag/MoO_3_	1.02	15.69	60	9.73	17.8	[128]
Islands	FTO/TiO_2_/MAPbI_3_ DVB (5%, molar fraction)/PTAA/PEDOT:PSS/ITO	0.98	16.52	73.79	11.95	-	[175]
Biomimetic	ITO/SnO_2_/Perovskite/Spiro-OMeTAD/MoO_3_/IZO	0.97	13.65	79.6	10.53	32.5	[176]
Nanopillar	FTO/c-TiO_2_/AAO + MAPbI_3−x_Cl_x_ (90–460 nm thick)/Spiro-OMeTAD/MoO_x_/ITO	1.03	17.72	72.38	13.27	26.3	[127]

## Figures and Tables

**Figure 1 nanomaterials-13-01084-f001:**
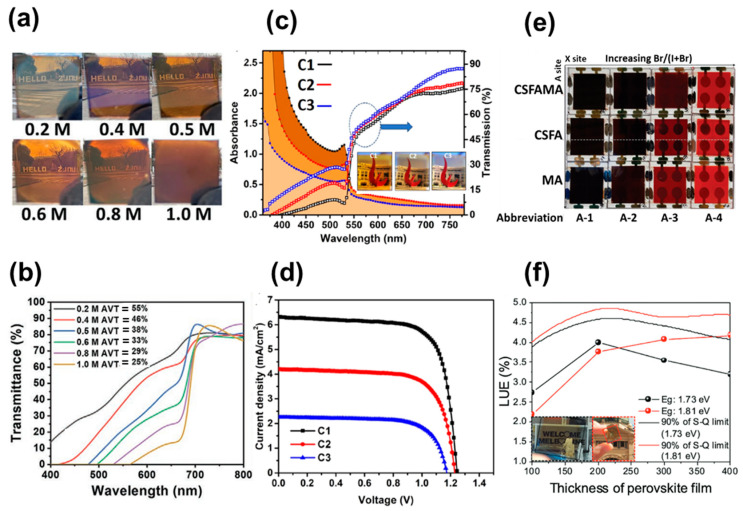
(**a**) Photographic images and (**b**) transmittance spectra of ST-PSCs for perovskite films with different concentrations. (**c**) UV-Visible absorption spectra and transmission spectra for device with FaPbBr_3_. (**d**) J-V curves with different concentrations. (**e**) Photograph of perovskite films’ perovskite compositions. (**f**) Dependence of LUE values on perovskite film thickness. Reprinted with permission from Refs. [25,27,28].

**Figure 3 nanomaterials-13-01084-f003:**
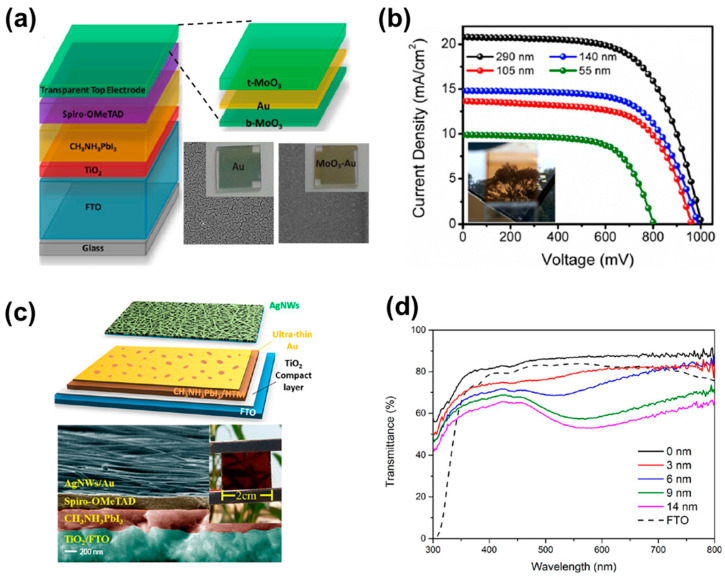
(**a**) Schematic representation of used PSC architecture. Enlarged view of the used top-electrode. SEM image of the used Au film, and of the modified electrode with MoO_3_/Au films. (**b**) Current–voltage characteristics of the fabricated cell with different transparencies. (**c**) Schematic image of devices fabricated with AgNWs composite electrode and ultra-thin Au layer (UTA). (**d**) Transmittance spectra of fabricated devices with different thicknesses. Reprinted with permission from Refs. [30,81].

**Figure 5 nanomaterials-13-01084-f005:**
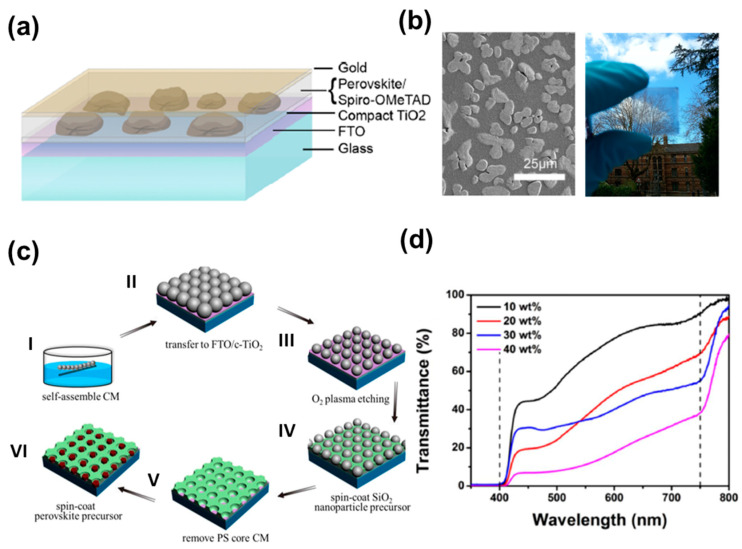
(**a**) Schematic representation of cells fabricated with dewetted region in planar perovskite heterojunction solar cells (**b**) FESEM and digital images of ST-PSC. (**c**) Schematic representation for the fabrication process for SiO_2_ HC structure patterned perovskite layer. (**d**) Transmittance spectra for these active layers of perovskite films. Reprinted with permission from refs. [20,122].

**Figure 6 nanomaterials-13-01084-f006:**
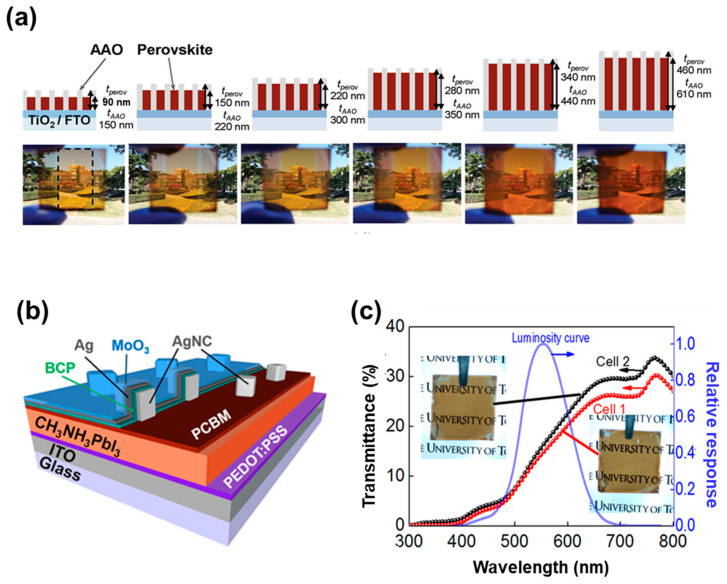
(**a**) Nanopillar-structured ST-PSC with different AAO templates and perovskite thicknesses. (**b**) Schematic representation of the device fabricated with Ag nanocubes. (**c**) Current–voltage spectra and transmittance spectra of plasmonic and non-plasmonic PVSCs together with the human luminosity curve. Reprinted with permission from Refs. [127,128].

**Figure 8 nanomaterials-13-01084-f008:**
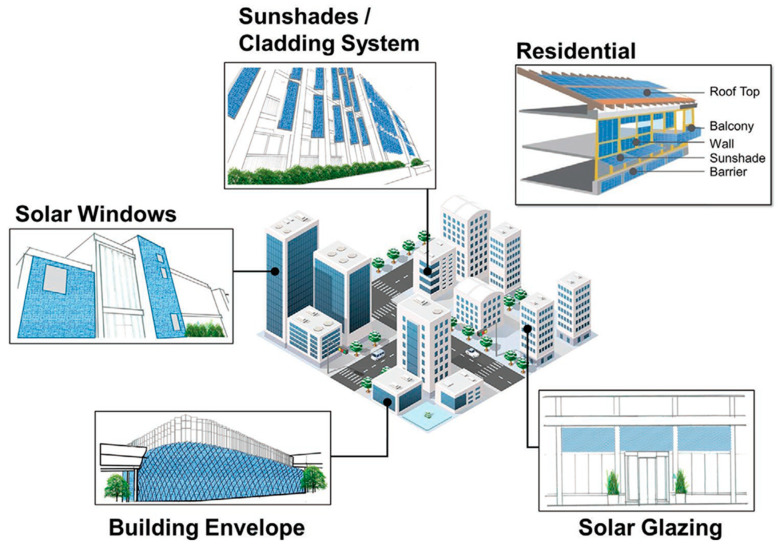
Application of BIPV into various surfaces of buildings unravelling the potential of solar energy for on-site energy production. Reprinted with permission from Ref. [158].

## Data Availability

The systematic review data used to support the findings of this study are included with the article.

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
