# Peer review of "Innovative Approaches to Semi-Transparent Perovskite Solar Cells"

_nanomaterials, 2023, doi:10.3390/nano13061084_

Round 1

Reviewer 1 Report

The power conversion efficiency (PCE) of perovskite solar cells (PSCs) have reached up to 25.7%, which is comparable with the silicon solar cells. The semi-transparent PSCs (ST-PSCs) are one promising application that utilizes the tunable transmittance of perovskite photoactive layers, which have attracted much attention. In this review, the authors summarized the innovative approaches to semi-transparent perovskite solar cells (ST-PSCs) including the development of perovskite photoactive layer, transparent electrodes, device structures and their applications. Overall, the manuscript is well written and organized. I am pleased to recommend this manuscript acceptable after well addressing the following issues.

1. In line 71 and 72, semi-transparent perovskite solar cells should be abbreviated as ST-PSCs. In line 580, transparent electrodes should be abbreviated as TEs.

2. The AVT is one of the main characteristics for semi-transparent devices. The wavelength range for the calculation of AVT should be stated, as disclosed in Adv. Funct. Mater. 2021, 31, 2107934, Sol. RRL 2022, 6, 2200308.

3. The flexible solar cells can provide scalability to various applications through integration with BIPV technology. It would be better to discuss more about the challenges and prospects of flexible devices.

Reviewer 2 Report

This review paper, “Innovative Approaches to Semi-Transparent Perovskite Solar Cells,” discusses the potential of semi-transparent perovskite solar cells (ST-PSCs) for use in tandem solar cells (TSC) and building-integrated photovoltaic (BIPV) applications. Although ST-PSCs have the advantage of tunable transmittance, achieving high efficiency while maintaining transparency is a challenge. The review covers the latest approaches to overcome this challenge, including band-gap tuning, high-performance charge transport layers and electrodes, and island-shaped microstructures. The paper also discusses the requirements and challenges in realizing ST-PSC and presents prospects for future developments. Overall, the paper provides a concise summary of the innovative approaches in ST-PSC and its potential for various applications. The topics are well-organized and thoroughly discussed. I would recommend it be published in Nanomaterials after the following minor issues are addressed:

  1. One crucial factor for practical PV applications is the stability of the materials and devices used. Therefore, it would be helpful to include a focused paragraph on stability in this review, providing guidance for the design and development of stable ST-PSCs. The current review touches upon various stability-related points, but they are not well-organized. A more structured and detailed discussion on stability would enhance the value of this review.
  2. The layout of this review could be improved for better readability. Additionally, the resolution of figures 2 and 7 should be revised to ensure that they are clear and easily understandable.

Reviewer 3 Report

The article, titled "Innovative Approaches to Semi-Transparent Perovskite Solar Cells", reviews a general and concise summary of the innovative approaches in ST-PSC, including advances in perovskite photoactive layer, transparent electrodes, device structures, and its applications in TSC and BIPV. However, there are some points, which should be clarity.

1.       The concluding introduction takes up too much of the article,but the explanation of the principle and mechanism of the method is too vague.

2.       As a review requires a lack of integrity, the author needs to link the contents of each chapter together as much as possible.

3.       The introduction can be improved. The articles related to some applications of solar cells should be added such as: Physical Chemistry Chemical Physics, 2022, 24, 4871 – 4880; Coatings 2022, 12(11), 1653; Energy Conversion and Management, 2022, 252, 115033; Coatings 2021, 11(7), 748;

4.       The reference should focus on principle analysis rather than simple statement work.

This manuscript can be considered for publication only when the above-mention questions were especially stressed in the revised manuscript. The referee would like to review a revised version of this paper in the future.

Round 2

Reviewer 3 Report

Accept in present form.